# RET Inhibitors in Non-Small-Cell Lung Cancer

**DOI:** 10.3390/cancers13174415

**Published:** 2021-09-01

**Authors:** Priscilla Cascetta, Vincenzo Sforza, Anna Manzo, Guido Carillio, Giuliano Palumbo, Giovanna Esposito, Agnese Montanino, Raffaele Costanzo, Claudia Sandomenico, Rossella De Cecio, Maria Carmela Piccirillo, Carmine La Manna, Giuseppe Totaro, Paolo Muto, Carmine Picone, Roberto Bianco, Nicola Normanno, Alessandro Morabito

**Affiliations:** 1Thoracic Medical Oncology, Istituto Nazionale Tumori IRCCS “Fondazione G. Pascale”, 80131 Napoli, Italy; priscillacascetta@gmail.com (P.C.); v.sforza@istitutotumori.na.it (V.S.); anna.manzo@istitutotumori.na.it (A.M.); giuliano.palumbo@yahoo.it (G.P.); espositogiovanna87@gmail.com (G.E.); a.montanino@istitutotumori.na.it (A.M.); r.costanzo@istitutotumori.na.it (R.C.); c.sandomenico@istitutotumori.na.it (C.S.); 2Department of Oncology and Hematology, Azienda Ospedaliera Pugliese-Ciaccio, 88100 Catanzaro, Italy; guidocarillio@gmail.com; 3Department of Pathology, Istituto Nazionale Tumori IRCCS “Fondazione G. Pascale”, 80131 Napoli, Italy; r.dececio@istitutotumori.na.it; 4Clinical Trials Unit, Istituto Nazionale Tumori IRCCS “Fondazione G. Pascale”, 80131 Napoli, Italy; m.piccirillo@istitutotumori.na.it; 5Thoracic Surgery, Istituto Nazionale Tumori IRCCS “Fondazione G. Pascale”, 80131 Napoli, Italy; c.lamanna@istitutotumori.na.it; 6Department of Radiotherapy, Istituto Nazionale Tumori IRCCS “Fondazione G. Pascale”, 80131 Napoli, Italy; g.totaro@istitutotumori.na.it (G.T.); p.muto@istitutotumori.na.it (P.M.); 7Department of Radiology, Istituto Nazionale Tumori IRCCS “Fondazione G. Pascale”, 80131 Napoli, Italy; c.picone@istitutotumori.na.it; 8Department of Clinical Medicine and Surgery, Oncology Division, University of Naples Federico II, 80131 Napoli, Italy; robianco@unina.it; 9Cellular Biology and Biotherapy and Scientific Directorate, Istituto Nazionale Tumori, “Fondazione G.Pascale” IRCCS, 80131 Napoli, Italy; n.normanno@istitutotumori.na.it

**Keywords:** NSCLC, RET, selpercatinib, pralsetinib, cabozantinib, vandetanib, lenvatinib

## Abstract

**Simple Summary:**

Non-small cell lung cancer (NSCLC) remains a significant cause of death worldwide, despite the significant progresses to date. Multiple molecular alterations have been identified in NSCLC, leading to the development of target-based agents that have shown significant clinical benefits. Rearranged during Transfection (RET) fusions have recently emerged as a new potential target and a number of non-selective and selective RET inhibitors have been tested in RET positive NSCLC. In this review we analyse and summarise the characteristics of RET functions and its alterations in NSCLC. We then present the state of the art RET inhibitors in the treatment of NSCLC, discussing the ongoing trials and the future perspectives for RET positive (RET+) NSCLC patients.

**Abstract:**

RET rearrangements are observed in 1–2% of non-small-cell lung cancer (NSCLC) patients and result in the constitutive activation of downstream pathways normally implied in cell proliferation, growth, differentiation and survival. In NSCLC patients, RET rearrangements have been associated with a history of non-smoking, a higher rate of brain metastasis at initial diagnosis and a low immune infiltrate. Traditionally, RET fusions are considered mutually exclusive with other oncogenic drivers, even though a co-occurrence with EGFR mutations and MET amplifications has been observed. Cabozantinib, vandetanib and lenvatinib are the first multi-kinase inhibitors tested in RET-rearranged NSCLC patients with contrasting results. More recently, two selective RET inhibitors, selpercatinib and pralsetinib, demonstrated higher efficacy rates and good tolerability and they were approved for the treatment of patients with metastatic RET fusion-positive NSCLC on the bases of the results of phase II studies. Two ongoing phase III clinical trials are currently comparing selpercatinib or pralsetinib to standard first line treatments and will definitively establish their efficacy in RET-positive NSCLC patients.

## 1. Introduction

The RET (Rearranged during Transfection) gene encodes a single-pass transmembrane receptor tyrosine kinase (RTK) physiologically involved in renal morphogenesis, neural and neuroendocrine tissue development as well as spermatogonial stem cell maintenance. The RET protein consists of an extracellular, a transmembrane and an intracellular region. The N-terminal extracellular region contains four highly repeated domains (cadherin-like domains) as well as a cysteine-rich domain and each domain is implied in both normal protein conformation and the construction of active ternary complexes [1]. The transmembrane domain links the extracellular cysteine-rich domain with the intracellular tyrosine kinase (TKI) domain, which ends with isoform-specific tails. In order to be activated, the RET protein requires the constitution of a heterogeneous ternary complex, which usually includes the glial cell line-derived neurotrophic factor (GDNF) family ligands and GDNF family co-receptors (GFRα1-4) (Figure 1). Once assembled, newly formed ternary complexes lead to the auto-phosphorylation of the RET TKI domains, resulting in the activation of downstream signalling pathways normally implied in cell proliferation, growth, differentiation and survival, such as RAS/MAPK, PI3K/AKT, PKC and JAK-STAT [2]. RET gain-of-function alterations have been identified in multiple solid tumours. By sequencing more than 10,000 different metastatic tumours, RET alterations have been found in 2.4% of all cases, primarily in thyroid cancers and NSCLC. Oncogenic RET gain-of-function alterations mainly occur via either germline/somatic mutations or chromosomal rearrangements. Point mutations represent the main RET alteration in medullary thyroid cancer and may occur in either the extracellular domain or the intracellular TKI domain, resulting in a ligand-independent RET activation. Instead, RET fusions are frequently found in NSCLC as well as in papillary thyroid cancer and could result in either ligand-independent activation or aberrant RET expression [3]. Based on previous data, both selective and non-selective RET inhibitors have been explored in NSCLC in multiple clinical trials. 

In this review we present the state of the art RET inhibitors in the treatment of NSCLC, discuss the ongoing trials and the future perspectives for RET positive (RET+) NSCLC patients and provide an updated panorama of this topic, especially on selective RET inhibitors.

### RET Rearrangements in NSCLC

RET rearrangements are rarely found in NSCLC patients (1–2%). In most cases, patients harboring RET alterations do not display any concurrent oncogenic driver. However, some small retrospective studies have reported MET amplification and EGFR mutations in this subset of patients [4,5]. Moreover, a large study also confirmed the co-existence of other genetic alterations in RET+ patients. By analysing 4871 different tumour samples, the co-occurrence of genetic abnormalities has been found in the majority of RET- altered patients (81.8%, 72/88 patients), the most common being TP-53 associated genes (59.1%, 52/88 patients), cell cycle-associated genes (39.8%, 35/88 patients), the PI3K signalling pathway (30.7%, 27/88 patients), MAPK effectors (22.7%, 20/88 patients) or other tyrosine kinase families (21.6%, 19/88 patients) such as FGFR families, EGFR, ALK, HER2, PDGFRα and PDGFRβ [6]. Finally, de novo RET fusions are a well described acquired mechanism of resistance to first, second and third generation anti-EGFR TKIs in EGFR mutant NSCLC patients [7,8]. To date, at least 12 fusion RET partner genes have been identified, the most common being KIF5B and CCDC6 (70–90% and 10–25% of all cases, respectively). Despite the breakpoints and fusion partners, a newly formed fusion gene occurs when a 5′ sequence encoding for a coiled coil domain of a RET fusion partner juxtaposes with the 3′ RET sequence encoding for the intracellular tyrosine kinase domain of RET, which usually preserves its functions. Coiled coil domains of these neo-constituted proteins induce ligand-independent homodimerization and the activation of the RET TKI domain by autophosphorylation, resulting in the constitutive stimulation of downstream signalling pathways [9]. RET rearrangements have been identified in tumour specimens via classic techniques, such as FISH and RT-PCR. Both these techniques, however, display some limits, namely their inability to detect de novo or unknown RET fusion partners. By analysing multiple genes at the same time, these limitations have been overcome by innovative techniques such as next generation sequencing (NGS). Moreover, NGS can also be used to investigate both RNA and DNA tumours and may therefore amplify the RET fusion detection rate [2]. Furthermore, NGS can be performed in both tumour specimens and in liquid biopsies, with obvious benefits in terms of invasiveness [10]. On the contrary, due to variable staining patterns as well as weak reactivity, immunohistochemistry (IHC) has been largely exceeded by the aforementioned techniques [11].

Traditionally, RET fusions have been related to young females with a history of non- smoking, although these clinical characteristics are still controversial and may differ between Asian and non-Asian populations [4,5,12,13,14]. Due to a higher percentage of RET fusions in NSCLC patients with a previous history of radiation (5.4% versus 0.4% of cases), gamma rays have been implied as a putative causal factor and these data were also validated in preclinical settings [15]. As previously mentioned, RET fusions are often present in NSCLC patients without other oncogenic drivers. Thus, the clinical and pathological characteristics of RET+ patients may differ from what has been observed for those with other oncogenic drivers. In 2012, Wang et al. analysed 936 tumour specimens obtained from Chinese patients who underwent surgery for mainly early-stage NSCLC. Although no statistical significance was achieved, RET+ patients tended to be younger than those harbouring EGFR mutations (RET+ vs. EGFR+ patients with <60 years: 72.7 % vs. 47.8%, *p* = 0.131). Likewise, RET+ tumours were poorly differentiated compared to what was observed for ALK positive, EGFR positive or RET negative (RET−) disease (poor differentiation in RET+ vs. ALK+: 63.6% vs. 25%, *p* = 0.029, poor differentiation in RET+ vs EGFR+: 63.6% vs. 23.8%, *p* = 0.007, poor differentiation in RET+ vs. RET−: 63.6% vs. 33.9%, *p* = 0.054). Interestingly, all RET+ patients had a small primary lesion (<3 cm) with a significantly higher percentage of N2 disease (54.5% for RET+ vs. 22.6% for other adenocarcinoma patients, *p* = 0.024) [12]. Nonetheless, these characteristics did not translate into worse clinical outcomes as no differences were seen in terms of either relapse free survival (RFS) or overall survival (OS) in RET+ versus RET-patients. Further data also confirmed the absence of prognostic impact in terms of PFS and OS for RET status both in Caucasian and in Asian NSCLC patients [4,14]. According to several studies, however, RET+ patients displayed a more advanced disease at the time of the initial diagnosis with up to 77% of patients presenting with stage III/IV versus 22% of patients presenting with stage I and II [5,13]. Furthermore, RET rearranged tumours have been linked to certain subtypes of adenocarcinoma, notably lepidic, solid and papillary, both in Asians and in non-Asians [4,13,16]. Interestingly, some initial reports showed that lymphangitic spread and psammoma bodies were frequently reported in a small series of RET-rearranged NSCLC, suggesting that RET assessment should be encouraged in those cases [17]. In an additional and larger retrospective study, however, no difference in the incidence of lymphangitic carcinomatosis was seen among RET+, ALK+ or ROS1+ advanced NSCLC patients. Compared with these other molecular subtypes, RET+ patients had a higher frequency of neuroendocrine histology (RET+ versus ALK+: 12% vs. 2%, *p* = 0.025; RET+ vs. ROS1+: 12% vs. 0%, *p* = 0.010) as well as peripheral primary tumours (RET+ vs. ALK+: 69% vs. 47%, *p* = 0.029; RET+ vs. ROS1: 69% vs. 36%, *p* = 0.003). Moreover, brain metastases were more likely to present at initial diagnosis in RET+ patients than in ROS1+ patients (RET+ vs. ROS1+: 32% vs. 10%; *p* = 0.039), whereas no differences were observed in RET+ and ALK+ patients (RET+ vs. ALK+: 32% vs. 25%; *p* = 0.592) [18]. Subsequent data also identified RET fusions as an independent risk factor for brain metastases in advanced NSCLC patients [19]. Finally, RET fusions have been related to a low response to immune checkpoint inhibitors due to a low TMB (tumour mutational burden) as well as a low level of PD-L1 expression [20].

## 2. Non-Selective RET Inhibitors

A number of multi-kinase inhibitors have been evaluated in RET-rearranged NSCLC, including cabozantinib, vandetanib and lenvatinib, with contrasting results (Table 1).

### 2.1. Cabozantinib

Cabozantinib is an oral multi-kinase inhibitor that is active against VEGFR2, MET, ROS1, AXL, KIT and TIE2, with decreased activity against RET (IC50 = 5.2 nM) [21]. Several clinical studies have demonstrated the efficacy of this drug in specific NSCLC cohorts, in particular in RET-rearranged lung cancer. After the identification of RET fusion in NSCLC by Ju et al. [22], Drilon et al. first reported the clinical activity of cabozantinib (60 mg/day) in three chemo-pretreated NSCLC patients harboring RET fusion [23]. Among them, two patients showed a partial response and a third patient had prolonged stable disease. Each patient who exhibited adverse events (AEs) required dose reduction thereafter (grade 3 proteinuria and hypertension). However, these toxicities were manageable with dose modifications and all patients manifested a prolonged response (responses or stable disease). Based on this evidence, an open-label single arm phase II trial was conducted to evaluate cabozantinib in both pretreated and naive RET+ NSCLC patients [24]. A total of 26 patients were enrolled and treated with cabozantinib (60 mg/day). Although no complete responses were observed, 28% of patients responded to cabozantinib. Moreover, responses to treatment were seen early, with a high percentage rate of tumour shrinkage (≥30% tumour reduction in 70% of patients). The median duration of the response was 7.0 months. The median PFS was 5.5 months (95% CI: 3.8–8.4) and the median OS was 9.9 months (95% CI: 8.1–not reached), respectively. Almost all patients (96.4%) experienced AEs, mainly with low grade toxicities (grade 1–2). The most common AEs were grade 3 elevated lipase, increased hepatic enzymes, a decreased platelet count and hypophosphataemia. The onset of grade 2 and 3 AEs required dose reduction in 73% of patients, the most common being palmar-plantar erythrodysaesthesia, fatigue and diarrhoea. 

Preclinical studies have suggested that EGFR signalling could play a central role in reducing RET inhibitors’ efficacy in NSCLC cell lines, thus providing a rationale for co-targeting both EGFR and RET in order to reduce the onset of drug resistances. By using NSCLC cell lines harbouring ALK, ROS1, RET and NTRK1 fusions, Vaishnavi et al. also confirmed that EGFR signalling was involved at different levels in determining the resistance to multi-kinase inhibitors so that treatment with gefitinib was able to abrogate EGFR contributions [25]. For this reason, a combination of erlotinib and cabozantinib was evaluated in a phase I–II trial [26]. This trial enrolled 54 pretreated NSCLC patients who received daily doses of cabozantinib plus erlotinib in a 3 + 3 design using combination doses across 5 cohorts in 2 parallel arms (A and B). Across all dose levels, 12 patients experienced at least 1 dose limiting toxicity (DLT): diarrhoea, elevated Aspartate Transaminase (AST), palmar-plantar erythrodysesthesia, mucositis, hypertension, hypokalemia, elevated lipase and fatigue. The most frequent grade 3–4 AEs were diarrhoea (26%), fatigue (15%), dyspnea (12%) and hypoxia (9%). The combination of these drugs was safe and encouraging clinical activity was observed in a largely erlotinib pretreated population, including patients with EGFR T790M and MET amplification. Based on these data, cabozantinib alone or with erlotinib was tested in a phase II trial to assess the improvement in PFS for cabozatinib over erlotinib in patients with EGFR wild-type NSCLC [27]. A total of 125 pretreated NSCLC patients without mutations in EGFR were enrolled and randomly assigned to one of the three treatment arms (cabozantinib alone 60 mg/day, cabozantinib 40 mg/day plus erlotinib 150 mg/day, erlotinib alone 150 mg/day). The primary endpoint was PFS. Secondary endpoints included OS, ORR and the toxicity associated with each regimen. Cabozantinib alone significantly increased PFS (median PFS 4.3 months, 95% CI: 3.6–7.4) compared to erlotinib (1.8 months, 95% CI: 1.7–2.2; HR 0.39, 80% CI: 0.27–0.55; *p* = 0.0003) and even the combination of cabozantinib plus erlotinib had a longer PFS (4.7 months; 95% CI: 2.4–7.4) compared to erlotinib alone (HR 0.37, 95% CI: 0.25–0.53; *p* = 0.0003). OS was also better with cabozantinib than with erlotinib (HR 0.68, 80% CI: 0.49–0.95; *p* = 0.071) and with cabozantinib plus erlotinib than with erlotinib alone (HR 0.51, 80% CI: 0.35–0.74; *p* = 0.011). The estimated median OS was 5.1 months (95% CI: 3.3–9.3) with erlotinib, 9.2 months (95% CI: 5.1–15.0) with cabozantinib and 13.3 months (95% CI: 7.6–not reached) with erlotinib plus cabozantinib. No differences were recorded in ORR between these three groups. Cabozantinib alone or with erlotinib was more toxic and was associated with an increased occurrence of grade 3 or worse AEs compared to erlotinib alone. Consistent with the literature, the most common grade 3 or 4 AEs were diarrhoea, hypertension, fatigue, oral mucositis and a thromboembolic event. Since testing for RET rearrangements was not mandatory in either of these two trials, the real efficacy of combining cabozantinib with anti-EGFR TKIs in RET+ NSCLC patients needs to be further elucidated. 

### 2.2. Vandetanib

Vandetanib is an oral multi-kinase inhibitor that selectively targets RET, VEGFR and EGFR signalling [28]. Phase I and phase II trials in various advanced tumours demonstrated that single agent vandetanib was well tolerated at a daily dose up to 300 mg.

The efficacy of vandetanib was tested in four phase III trials as single agents or in combination with chemotherapy for patients with advanced NSCLC unselected for RET rearrangements [29,30,31,32]. In the ZEST study, 1240 chemo-pretreated NSCLC patients were randomly assigned to either vandetanib (300 mg daily) or erlotinb (150 mg daily) as second or third line treatment [29]. No significant improvement in PFS was observed for vandetanib over erlotinib (HR 0.98; 95.22% CI: 0.87–1.10; *p* = 0.721). No significant differences were seen in terms of OS (HR 1.01; *p* = 0.830), ORR (12% for both arms) and time to deterioration of symptoms. The most frequent AEs with vandetanib were diarrhoea (50% vs. 38%) and hypertension (16% vs. 2%). Grade ≥ 3 toxicities were more frequent with vandetanib than with erlotinib (50% vs. 40%). Moreover, the ZEPHYR trial aimed to compare vandetanib (300mg/day) with a placebo in patients whose disease progressed after at least two lines of treatment, including an EGFR TKI inhibitor [30]. Vandetanib performed better than the placebos in terms of PFS (HR 0.63, *p* < 0.001) and ORR (2.6% vs. 0.7%, respectively). However, vandetanib did not significantly increase OS compared with the placebo (8.5 vs. 7.8 months; HR 0.95; 95.2% CI: 0.81–1.11; *p* = 0.527). Common adverse events in the vandetanib arm were diarrhoea (46% vs. 11%), rash (42% vs. 11%) and hypertension (26% vs. 3%). In the ZODIAC study, 1391 patients with advanced NSCLC progression after first line chemotherapy were enrolled to receive second line docetaxel plus vandetanib (100 mg/day) or docetaxel plus the placebo [31]. The median PFS was significantly improved by adding vandetanib to docetaxel (4.0 vs. 3.2 months; HR (hazard ratio) 0.79; 97.58% CI: 0.70–0.90; *p* < 0.0001). The advantage was seen across all subgroups, including women (PFS 4.6 vs. 4.2 months, HR 0.79; 97.58% CI: 0.62–1.00, *p* = 0.024). However, there was no significant difference in OS (10.3 vs. 9.9 months; HR 0.95, 95% CI: 0.84–1.07; *p* = 0.371). Rash (9% vs. 1%), neutropenia (29% vs. 24%), leukopenia (14% vs. 11%) and febrile neutropenia (9% vs. 7%) were the most common grade ≥3 AEs in the docetaxel plus vandetanib arm. Similarly, the ZEAL trial compared pemetrexed plus vandetanib (100 mg/day) with pemetrexed plus placebo as a second line treatment for 534 NSCLC patients, 21% of whom had squamous histology [32]. No significant advantage in terms of PFS was noted by adding vandetanib to chemotherapy. However, a trend towards a better PFS (HR 0.86; 97.58% CI: 0.69–1.06; *p* = 0.108) and OS (HR 0.86; 97.54% CI: 0.65–1.13; *p* = 0.219) was noted, with a similar advantage observed for females. There was a statistically significant improvement in ORR (19.1% vs. 7.9%, *p* < 0.001) and time to deterioration of symptoms (HR 0.61, *p* = 0.004). Common AEs were rash (38% vs. 26%), diarrhoea (26% vs. 18%) and hypertension (12% vs. 3%). The evidence suggested that adding vandetanib to pemetrexed results in a lower rate of chemo-induced toxicities: anaemia 8% vs. 22%, nausea 29% vs. 37%, vomiting 15% vs. 22%, fatigue 37% v.s 45% and asthenia 11% vs. 17%. The incidence of QT prolongation was <1%. There was no increase in bleeding or thrombotic events in the vandetanib arm. As previously highlighted, however, patients included in these trials were not selected for RET rearrangements. Thus, no data regarding this subgroup of patients can be extrapolated. Indeed, further trials tested vandetanib in specifically selected RET+ patients. In an open-label phase II trial by Lee SH et al., 18 patients with metastatic or recurrent NSCLC harbouring RET rearrangements confirmed by fluorescence in situ hybridization were enrolled [33]. Most of the patients received two or more chemotherapy regimens. ORR was observed in 18% of patients and a stable disease was seen in 47% of the population, with a disease control rate (DCR) of 65%. Moreover, vandetanib showed a PFS of 4.5 months and an OS of 11.6 months after a median follow up of 14 months. The safety profile was consistent with the previous studies. Another phase II trial by Yoh K et al. (LURET) screened 1536 patients with EGFR mutation-negative NSCLC and discovered 34 (2%) RET-rearranged cases [34]. Among 19 patients receiving 300 mg/day of vandetanib, 47% achieved an objective response and the median PFS was 4.7 months. The most common grade 3 or grade 4 AEs were hypertension (58%), diarrhoea (11%), rash (16%), dry skin (5%) and QT prolongation (11%).

### 2.3. Lenvatinib

Lenvatinib is a multi-kinase inhibitor of RET, KIT, VEGFR1–3, PDGFRα and FGFR1–4. It is currently used for radioactive iodine-refractory differentiated thyroid cancer in combination with everolimus for patients with advanced renal cell carcinoma, for the first line treatment of patients with unresectable hepatocellular carcinoma and in combination with pembrolizumab for the treatment of patients with advanced endometrial carcinoma [35]. In a double-blind, placebo-controlled, randomised phase II study that enrolled 134 non-squamous, heavily pretreated (≥3 lines) NSCLC patients unselected for RET rearrangements, lenvatinib (24 mg once daily) plus best supportive care (BSC) improved OS (38.4 vs. 24.1 weeks, *p* = 0.065), PFS (20.9 vs. 7.9 weeks, *p* < 0.001) and ORR (10.1% vs. 2.2%) compared with BSC [36]. In a phase II trial conducted in 25 pretreated patients with RET+ lung adenocarcinoma, lenvatinib (24 mg once daily) showed 16% in ORR (95% CI: 4.5–36.1), with no significant differences between patients with the KIF5B–RET fusion variant and the CCDC6–RET fusion variant [37]. The median PFS was 7.3 months (95% CI: 3.6–10.2) and it was longer in patients with the KIF5B–RET fusion variant versus the CCDC6–RET fusion (9.1 months vs. 3.6 months, respectively). The safety profile of Lenvatinib was manageable with hypertension, nausea, diarrhoea and proteinuria being the most common AEs. A phase Ib/II trial was conducted with lenvatinib (20 mg/day) plus pembrolizumab (200 mg every 3 weeks) in pretreated patients with different tumour types (21 patients with advanced NSCLC) who were not previously selected for PDL1 or other biomarkers. In NSCLC patients, the overall ORR was from 14.6% to 57.0% and the median PFS was 5.9 months (95% CI, 2.3 to 13.8 months) [38].
cancers-13-04415-t001_Table 1Table 1Non-selective RET inhibitors in RET+ NSCLC patients.AuthorRegimenSettingPtsORR (%)Median PFS (Months)Median OS (Months)Drilon, A. et al., 2016 [24]Cabozantinib 60 mg/dayPretreated or untreated26285.5 9.9Neal, J.W. et al., 2016 [27]Cabozantinib 60 mg/day vs. cabozantinib 40 mg/day + erlotinib150 mg/day Vs. erlotinib 150 mg/dayPretreated12511 vs. 3 vs. 34.3 vs. 4.7 vs. 1.8 9.2 vs. 13.3 vs. 5.1 Lee, S.H. et al., 2017 [33]Vandetanib 300 mg/dayPretreated18(17 evaluable)184.511.6Yoh, K, et al., 2017 [34]Vandetanib 300 mg/dayPretreated19474.711.1Hida, T, et al., 2019 [37]Lenvatinib 24 mg/dayPretreated2516%7.3-ORR: objective response rate; PFS: progression-free survival; OS: overall survival; Pts: patients.

## 3. Selective RET Inhibitors

Small, highly selective RET inhibitors have been developed with the aim of overcoming treatment-related toxicities commonly seen with non-selective RET inhibitors [39,40]. Among these, selpercatinib and pralsetinib received FDA approval for the treatment of NSCLC harbouring RET alterations (Table 2).

### 3.1. Selpercatinib

Selpercatinib (LOXO-292) is an oral TKI inhibitor with potent and specific activity against the RET kinase domain, including multiple RET alterations such as fusions, activating point mutations and predicted acquired resistance mutations. Its activity on kinases other than RET is negligible. As one of the most selective RET inhibitors, selpercatinib represents a step forward for the management of RET+ lung cancer patients who have been traditionally treated with standard of care therapies [39,40]. The clinical safety of selpercatinib and its activity profile have been tested in the phase I–II open-label, first-in-human, clinical trial LIBRETTO-001. The study enrolled, in separate cohorts, patients with advanced or metastatic RET+ NSCLC who had disease progression after platinum-based chemotherapy and patients with the same biological characteristics but who were treatment naïve [41]. In the phase I dose escalation, nine dose levels ranging from 20 mg once daily (QD) to 240 mg twice a day (BID) were investigated. At the 240 mg BID dose level, two DLTs were reported: one grade 3 tumour lysis syndrome and one grade 3 thrombocytopenia [42]. The recommended phase II dose established at 160 mg BID demonstrated a favorable safety profile and durable antitumour activity. In phase II, 105 patients were enrolled with RET+ NSCLC who were pretreated with platinum chemotherapy. The median age was 61 years (range, 23–81) and the ECOG performance status was 0–1 (98%) or 2 (2%). Although many of these patients were heavily pretreated, with a median of three prior lines of systemic therapies (including immunotherapy and multitargeted kinase inhibitors with anti-RET activity), an ORR of 64% (95% CI: 54%–73%) was observed, with a median duration of response of 17.5 months (95% CI: 12–NE months; NE, not estimable) as determined by the independent review committee. The responses were observed regardless of the RET fusion partner. At one year, 66% (95% CI: 55 to 74) of all patients were progression-free and the median PFS was 16.5 months (95% CI: 13.7 to NE). It is important to note that 55% of patients received a previous treatment with anti-PD-1/PD-L1 either sequentially or concurrently with platinum-based chemotherapy, achieving an ORR of 66% (95% CI: 52%–78%) with a median duration of response of 12.5 months (95% CI: 8.3-NE). Given the high risk for patients with RET-alterations of developing brain metastasis [43], selpercatinib was also designed to achieve a significant central nervous system (CNS) penetration and activity; thus, 11 patients with measurable brain metastasis were enrolled and responses in the intracranial lesions were observed in 10 of them, with a duration of response of 10.1 months (95% CI: 6.7-NE). The major benefit was observed in the cohort of 39 treatment-naive patients: the ORR was 85% (95% CI: 70–94%) and, to date, the median duration of response and PFS have not been reached. Regarding the safety profile, selpercatinib was well-tolerated and clinically manageable, with lower rates (2%) of study drug discontinuation due to AEs, such as an increase in the alanine aminotransferase level (in 2 patients) and drug hypersensitivity (in 2 patients). Grade 3 or 4 treatment-related adverse events observed in ≥5% of cases were hypertension (in 14% of the patients), increased blood levels of transaminases (alanine aminotransferase in 13%), (aspartate aminotransferase in 10%), hyponatremia and lymphopenia (6%, both). A total of 6 grade 5 adverse events were reported and considered unrelated to the study treatment by the investigators, including sepsis and cardiac arrest, multiple organ dysfunction syndrome, pneumonia and respiratory failure. Dose reduction was necessary in 30% of patients because of treatment-related adverse events. On the basis of these data, on May 8th of 2020, the US Food and Drug Administration (FDA) granted an accelerated approval of selpercatinib for the treatment of adult patients with metastatic RET+ NSCLC regardless of the line of therapy.

### 3.2. Pralsetinib

Pralsetinib (BLU-667) is a small molecule that strongly inhibits the RET kinase domain. In vitro studies demonstrated that, compared with cabozantinib and vandetanib, this molecule is 8 to 28-fold more potent against the wild-type RET kinase domain. Moreover, pralsetinib also displays a strong activity against common oncogenic RET alterations, such as RET M918T, KIF5B–RET and CCDC6–RET fusions. Pralsetinib also has an 88-fold higher selectivity against RET over VEGFR2 compared to what was observed with other multi-kinase inhibitors [44]. Furthermore, pralsetinib is also able to overcome acquired resistance to multi-kinases inhibitors as well as to third generation anti-EGFR TKIs such as osimertinib [7,44]. Pralsetinib is currently being evaluated in the multicenter phase I-II ARROW trial. This trial consists of a dose escalation phase and a subsequent dose expansion phase. In the dose escalation phase, patients with advanced RET-altered solid tumours have been enrolled in order to find the recommended dosage of the drug (400 mg daily). Instead, the currently ongoing dose expansion phase aims to assess the efficacy of pralsetinib in seven different cohorts, including patients with advanced RET+ NSCLC either as first line or in subsequent lines of treatment. Significant key eligibility criteria include no other additional driver mutations as well as ECOG PS 0–1. Importantly, patients with asymptomatic brain metastasis were allowed to enter the trial. The primary endpoints are safety and ORR evaluated by a blinded independent central review as per the RECIST 1.1 criteria. The first results regarding the RET+ NSCLC cohort were presented at the 2019 ASCO Annual Meeting. A total of 79 patients were enrolled, the majority of whom were highly pretreated with two prior therapies as a median number, primarily chemotherapy (76%), immunotherapy (41%) and multi kinase inhibitors (27%). Brain metastasis at the baseline occurred in 39% of patients. The most common RET -fusion partner was KIF5B (44/79 of cases), followed by CCDC6 (16/79 cases). In 19 patients, the RET fusion partner remained unknown. The efficacy population included 57 patients, all with at least one follow-up assessment. The ORR was observed in 56% of cases and 6 patients manifested controlled disease for more than 6 months. DCR was observed in 91% of patients (52/57 patients). Responses occurred regardless of the number and type of prior therapies as well as RET fusion partners, although patients previously treated with platinum agents displayed a more significant ORR (60%). It was also demonstrated that pralsetinib had significant intracranial activity. In terms of side effects, pralsetinib has been well tolerated with mainly low grade toxicities (28% had ≥ grade 3 events). The most commonly observed adverse events were AST and ALT increase (22% and 17%, respectively), hypertension (18%), constipation (17%), neutropenia (15%) and fatigue (15%) [45]. More importantly, after 8 weeks of treatment, patients treated with pralsetinib experimented a major clearance in RET ctDNA. This phenomenon corresponded to either a partial response or stable disease at subsequent assessments [46]. The updated results were presented at the 2020 ASCO Annual Meeting, substantially confirming the efficacy and safety of pralsetinib. In 116 NSCLC patients, ORR overall was 65%. Interestingly, pralsetinib performed better in naïve patients than in platinum pretreated patients in terms of response rate, with an ORR of 73% versus 61%, respectively. On the contrary, DCR was higher in platinum pretreated patients than in naïve patients (DCR 95% versus 88% in platinum pretreated versus naïve patients, respectively). After a follow-up of 8.8 months, the median time to response was 1.8 months and the median duration of response was not reached. Indeed, no data on PFS are currently available [47]. Among eight patients with measurable central nervous system metastases at the baseline, an intracranial response was observed in four patients (complete response in two). Intracranial responses were long lasting, without progression after 6 months. The pharmacokinetic proprieties of pralsetinib suggest that this drug is primarily metabolised by cytochromes (mainly CYP3A4) and largely excreted in feces. No differences in the pharmacokinetic proprieties have been found according to age, sex or mild or moderate renal or hepatic impairment. Since food intake may alter drug absorption, the recommended dosage of pralsetinib is 400 mg orally taken once daily on an empty stomach [48]. Even in the absence of larger studies and given the promising efficacy data from the ARROW trial, the FDA recently approved pralsetinib for the treatment of RET+ advanced NSCLC patients.
cancers-13-04415-t002_Table 2Table 2Selective RET inhibitors in RET+ NSCLC patients.AuthorPhaseRegimenSettingPtsORR (%)Median PFS (Months)Drilon, A. et al., 2020 [41]I–IISelpercatinib 160 mg twice dailyPlatinum pretreated10564 (95% CI: 54–73%)16.5 (95% CI: 17.7–n.r.)IISelpercatinib 160 mg twice dailyUntreated3985 (95% CI: 70–94%)n.r. (95% CI: 13.8–n.r.)Gainor, J.F. et al., 2020 [47]I–IIPralsetinib 400 mg dailyPretreated8061 (95% CI: 50–72)-IIPralsetinib 400 mg dailyUntreated2673 (95% CI: 52–88)-n.r.: not reached; ORR: objective response rate; PFS: progression-free survival; OS: overall survival; IC: interval confidence; Pts: patients.

## 4. Discussion

RET fusions were recently identified in lung cancer and to date several phase II trials have investigated the role of RET non-selective multi-kinase inhibitors in RET+ lung cancer patients, with unsatisfactory clinical results. The weaknesses of these drugs was essentially due to their poor anti-RET potency and to their off-target side-effects, resulting in limited clinical activity burdened by excessive toxicities [39]. On the contrary, promising results have been reported to date in phase I–II studies with two selective RET inhibitors, selpercatinib and pralsetinib, with ORR in 56–85% patients, pretreated or not with multiple lines of chemotherapy and a good safety profile. Based on the aforementioned results, the FDA have granted an accelerated approval of selpercatinib and pralsetinib for the treatment of adult patients with metastatic RET+ NSCLC. However, the clinical research on RET inhibitors in NSCLC is just beginning and a number of relevant questions remain to be addressed: what is the best RET inhibitor for patients with advanced NSCLC and what will be their impact on overall survival? Are there alternative strategies to improve these results, such as combining anti-RET TKI with chemotherapy, immunotherapy or other TKIs? How is it possible to face the resistance mechanisms for RET inhibitors? 

For the first question, there are currently two ongoing phase III clinical trials that are comparing selpercatinib or pralsetinib to platinum-based chemotherapy with or without pembrolizumab as an initial treatment for advanced or metastatic RET+ NSCLC patients (Table 3). The LIBRETTO-431 trial is a phase III trial involving metastatic or stage IIIB–C naive patients not suitable for radical surgery or radiation therapy [49]; patients must have a RET gene fusion found from a tumour biopsy or in blood samples. In this trial, patients will be randomised to receive selpercatinib or pemetrexed plus carboplatin or cisplatin with or without pembrolizumab. The primary outcome measure is PFS. The results are expected in August 2025. The AcceleRET trial is a phase III trial involving metastatic or stage IIIB–C naive patients not suitable for radical surgery or radiation therapy with a RET gene fusion [50]. Patients will be randomised to receive pralsetinib or, if they are non-squamous patients, platinum plus pemetrexed with or without pembrolizumab, while if they are squamous patients, platinum plus gemcitabine. The primary outcome measure is PFS. The estimated enrollment is 250 patients. The results are expected in December of 2024. Overall, both these studies share a similar design and will clarify whether upfront anti-RET treatments should be preferred over chemotherapy alone or combined with pembrolizumab. Instead, no direct comparison between the efficacy of pralsetinib and selpercatinib could be made. 

Precision medicine could also be a helpful tool to further assess which is the best upfront RET inhibitor and how to improve its efficacy. Other RET inhibitors in clinical development are alectinib and brigatinib. Alectinib is under evaluation in the first line setting in the B-FAST trial, a multi-cohort phase II/III trial with an innovative diagnostic strategy based on a liquid biopsy to define different cohorts of treatment [51]. Cohort B is devoted to RET+ patients treated with alectinib at the dose of 1200 mg BID. The primary end point is the objective response rate. The results are expected in the last quarter of 2021. In the second line setting, the ROME trial is a proof-of-concept phase II trial which is trying to assess the efficacy of a treatment based on the genomic profile evidenced through Foundation One. This trial involves patients with different solid tumours, including NSCLC [52]. Enrolled patients must have progressed after at least one prior line of therapy. Next, the Foundation One profiling is performed and patients are therefore treated according to the mutation eventually found, independently from their type of cancers. For RET-mutated patients, the drugs that are allowed are alectinib and brigatinib. The primary end point is ORR. The estimated enrollment is 384 patients. The results are expected in August 2024. A section of the LUNG-MAP clinical trial is dedicated to patients with RET fusion. LUNG-MAP is an umbrella trial that has enrolled stage IV or unresectable lung cancer patients who have undergone at least first line treatment with platinum or, if they have received a treatment with platinum while in stage I–III, then progression must have happened in less than one year [53]. In the prescreening phase, patients are tested with Foundation One to assess or confirm the presence of the RET mutation and then they are treated with selpercatinib. The primary outcome measure is the response rate. The estimated enrollment is 124 participants. The results are expected in March 2023. A number of phase II trials are exploring the activity of other RET inhibitors in pretreated patients, including cabozantinib in the Creta trial and alectinib in the Alert-lung trial [54,55]. Even though these trials are designed to assess the efficacy of anti-RET TKIs in a selected population, they will also demonstrate to what extent precision medicine techniques should be considered helpful in case of doubts. Despite the high reliability, some discrepancies might arise when comparing liquid and tissue biopsies, mainly due to high intra-tumour heterogeneity. A possible implication of this concept that could represent a further investigational area would be the analysis of the tumour heterogeneity among upfront non-responsive patients.

For the second question, multiple ongoing trials are currently evaluating whether combined treatment strategies could lead to better results. Indeed, a phase II trial is exploring the activity of cabozantinib alone, cabozantinib plus nivolumab or cabozantinib plus nivolumab and ipilimumab in recurring stage IV NSCLC [56]. In this trial, arm T is devoted to patients with ROS1, MET and also RET fusions: in this arm, all patients will receive intravenous nivolumab every 28 days plus cabozantinib daily per os. Patients may have been treated with multiple lines, including biological therapy. The primary end point is PFS. The estimated enrollment in the whole trial is 169 patients. The results are expected in 2022. A phase III trial is ongoing to assess the efficacy of a novel multi-targeting tyrosine kinase inhibitor, anlotinib, in combination with chemotherapy in squamous NSCLC unselected patients for RET fusions. Patients can be enrolled only if disease progression has occurred for more than 12 months after the end of the last treatment and they will be randomised to receive anlotinib plus carboplatin and taxol or placebo plus carboplatin and taxol [57]. The primary outcome measure is PFS. The estimated enrollment is 386 patients. The results are expected in July 2022. Moreover, another phase III trial is assessing the efficacy of the combination of lenvatinib plus pembrolizumab versus pembrolizumab alone in the first line in patients with advanced NSCLC, a PDL-1 greater than or equal to 1%, but who have been selected for RET positivity [58]. As previously described, concurrent genetic alterations have been identified in tumour samples of RET-rearranged NSCLC patients, mainly in TP-53 associated genes, cell-cycle associated genes and PI3K signalling pathway [6]. Furthermore, both in vitro and in vivo models of RET+ NSCLC cell lines demonstrated a strong synergistic effect when non-selective anti-RET TKIs were combined with either CDK 4/6 inhibitors or PI3K/mTOR inhibitors [59]. In this scenario, available preclinical data have highlighted a potential role of multi-targeted agents in the treatment of RET-rearranged NSCLC, which is also going to be further verified in clinic. In an ongoing phase I trial including 13 RET+ NSCLC patients, vandetanib plus everolimus led to 54% of ORR with important responses also seen in brain metastases. The median PFS was 4.4 months. The major toxicities included diarrhoea (21%), thrombocytopenia (16%), QTc prolongation (5%) and rash (5%) and 17/19 patients required dose reduction because of toxicities [60]. 

Taken together, all these data seem to demonstrate a possible crosstalk between different pathways both in vivo and in vitro. However, given the predominant role of new selective anti-RET molecules, all these data might now be considered outdated. Thus, an interesting future field of investigation would be the further exploration of an upfront combination strategy using either pralsetinib or selpercatinib.

For the third question, despite the encouraging preliminary results obtained, the benefit from RET-selective inhibitors might be limited by the development of acquired resistance; hence, understanding and developing therapeutic strategies to overcome them is of primary interest. Solomon et al. firstly described a RET G810 solvent front mutation as a mechanism of resistance to selpercatinib in five patients with RET fusion-positive NSCLC and RET-mutant medullary thyroid cancer [61]. Thereafter, Lin and co. reported a multi-institutional analysis of repeat tumour or plasma biopsies from RET+ NSCLC patients treated selpercatinib or pralsetinib, highlighting the role of RET solvent front mutations G810C and G810S as on-target mechanisms of resistance and MET and KRAS amplification as an RET-independent mechanism of escape [62]. Finally, in a recently published paper, Rosen et al. demonstrated through single patient protocols that combination treatment with selpercatinib and crizotinib is safe and clinically active to overcome MET-amplification as a mechanism of resistance during treatment with selpercatinib [63]. A phase I-II trial is investigating the role of TPX-0046, a new potent RET inhibitor, that has shown activity even in tumour models with Solvent Front Mutations [64]. This trial enrolls patients with solid tumours harboring RET fusions in progression after previous therapy or who are ineligible or unlikely to benefit from standard treatment. Cohorts one and two are dedicated to NSCLC patients, for naive and pre-treated patients respectively. All patients will receive TPX-0046. ORR is the primary end point of phase II. The estimated enrollment is 362 patients. The results are expected in March 2025. To date, however, less is known about acquired resistance mechanisms to anti-RET TKIs and how to treat patient who progressed after these molecules. Therefore, this research area needs to be better characterised.

Currently, RET rearrangements in NSCLC represent an evolving topic with a growing number of manuscripts becoming available. In a similar review published by Choudhury and Drilon in 2020, the authors explained RET protein structure and its activation, but the topic of RET diagnostic techniques and concurrent mutations was not discussed. Furthermore, Choudhury et al. discussed the characteristics of RET+ NSCLC patients, but we tried to highlight the differences in terms of clinical and pathological presentation at diagnosis for this subgroup of patients. Despite similar trials cited in both these papers, our review provides an updated panorama of this topic, especially if considered selective RET inhibitors [65].
cancers-13-04415-t003_Table 3Table 3Ongoing trials with RET inhibitors in RET positive NSCLC patients.TrialPhaseSettingStagePtsTreatmentPrimary End Points**NCT04194944 (LIBRETTO-431)** [49]Phase IIIFirst lineStage IV or IIIB-C *250Selpercatinib vs. platinum + pemetrexed with or without pembrolizumabPFS**NCT04222972****(ACCELE-RET)** [50]Phase IIIFirst lineStage IV or IIIB-C *250Pralsetinib vs. platinum + pemetrexed with or without pembrolizumab (if non squamous) or platinum + gemcitabinePFS**NCT03178552****(B-FAST)** [51]Phase I/IIFirst lineStage IV or stage III * 50AlectinibORR**NCT04591431****ROME** [52]Phase IISecond lineStage IV384Alectinib or brigatinibORR**NCT04268550 (LUNG-MAP)** [53]Phase IISecond or subsequent linesStage IV or stage III *124SelpercetinibORR**NCT04131543 (CRETA)** [54]Phase IISecond or subsequent linesStage IV or stage III *25CabozantinibORR**NCT03445000 (ALERT-LUNG)** [55]Phase II Second or subsequent linesStage IV or stage III *44AlectinibORR**NCT03468985** [56]Phase IIPretreatedStage IV169Nivolumab + cabozantinibPFS**NCT04161391** [64]Phase I-IINaive or pretreatedStage IV362TPX-0046ORR* not suitable for radical surgery or radiation therapy; PFS: progression-free survival; ORR: objective response rate.

## 5. Conclusions

Selective RET inhibitors including selpercatinib and pralsetinib have been recently approved by the FDA for the treatment of RET+ advanced NSCLC patients on the basis of the positive results of phase II studies. Two ongoing randomised phase III studies are currently comparing these two drugs with standard first line treatment of patients with advanced NSCLC harbouring RET rearrangements and should better define their role in the first line setting. Open issues are the evaluation of the role of combining RET inhibitors with chemotherapy or immunotherapy to improve activity and to face resistance mechanisms of RET inhibitors. 

## Figures and Tables

**Figure 1 cancers-13-04415-f001:**
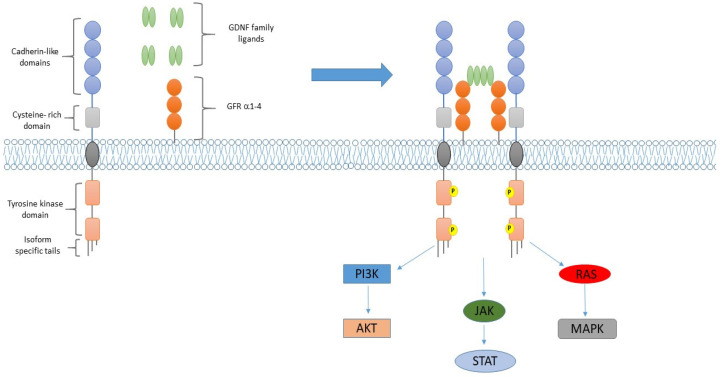
RET structure and its activation. The RET protein includes an extracellular, a transmembrane and an intracellular region. The RET extracellular region consists of four highly repeated domains (cadherin-like domains) as well as a cysteine-rich domain. The transmembrane domain links the extracellular region to the intracellular tyrosine kinase domain (TKI), which ends with isoform specific tails (on the left). Glial cell line-derived neurotrophic factor (GDNF) family ligands and GDNF family co-receptors (GFR α1-4) play a central role in RET activation. The binding of the GDNF ligand to the GFR co-receptor determines the construction of a ternary complex which includes RET, the GDNF ligand and the GFR co-receptors. The newly formed ternary complex leads to RET TKI domain phosphorylation. Phosphorylated RET TKI domains then activate the downstream signalling pathways implied in cell proliferation, growth, differentiation and survival such as RAS/MAPK, PI3K/AKT, PKC and JAK-STAT (on the right).

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
