# Peer review of "RET Inhibitors in Non-Small-Cell Lung Cancer"

_cancers, 2021, doi:10.3390/cancers13174415_

Round 1

Reviewer 1 Report

Review of “RET inhibitors in Non-Small-Cell Lung Cancer”

Concerns:

The review is well drafted and interesting. The main weakness is a similar review from 2020 titled "Decade in review: a new era for RET-rearranged lung cancers". It will be essential that the authors include this review in the discussion and justifies in the introduction the added value of the author's review. Comparing the two reviews, a few inconsistencies appeared in the number of patients and ORR reported (comparing table 1 in the two reviews). Furthermore, each of the two reviews list an additional clinical trial that were not listed in the other. The authors clearly must add extra value to obtain a review that is distinct from Drilon 2020. It is worth highlighting that the authors raise unresolved questions in the discussion. However, although the authors list ongoing investigation attempting to answer these questions, they could be more critical and also propose ideas of their own. Furthermore, a section on the role of preclincial models and how the authors see that preclincial models could accelerate the development of a more optimal treatment of patients with RET gene alterations could be a way to improve and distinguish the authors review. The review has sections with many clinical trial numbers in the main text which is daunting to read and better suited for tables. This section could become more readable by providing more discussion on the meaning and critical assessment of the listed numbers in the main text.

Minor:

L.100-102 is repeated text from abstract

L.188: AU: "Based on these evidences," → evidence is uncountable

L.201: AST not defined

Author Response

  • The review is well drafted and interesting. The main weakness is a similar review from 2020 titled "Decade in review: a new era for RET-rearranged lung cancers". It will be essential that the authors include this review in the discussion and justifies in the introduction the added value of the author's review.

Answer: we thank the reviewer for the positive comments and the suggestions. We justified at the end of the introduction the added value of our review: “In this review we present the state of the art of RET inhibitors in the treatment of NSCLC, discussing the ongoing trials and the future perspectives for RET positive (RET+) NSCLC patients, providing an updated panorama of this topic, especially on selective RET inhibitors”. In the discussion, we highlighted similarities and differences between our paper and "Decade in review: a new era for RET-rearranged lung cancers", as follow: “In conclusion, RET rearrangements in NSCLC represent an evolving topic with a growing number of manuscripts becoming available. In a similar review published by Choudhury and Drilon in 2020, the authors explained RET protein structure and its activation, but the topic of RET diagnostic techniques and concurrent mutations was not discussed. Furthermore, Choudhury et al. discussed the characteristics of RET + NSCLC patients, but we tried to highlight the differences in terms of clinical and pathological presentation at diagnosis for this subgroup of patients. Despite similar trials are cited in both these papers, our review provide an updated panorama of this topic, especially if considered selective RET inhibitors”.

  • Comparing the two reviews, a few inconsistencies appeared in the number of patients and ORR reported (comparing table 1 in the two reviews). Furthermore, each of the two reviews list an additional clinical trial that were not listed in the other.

Answer: we thank the reviewer for this comment. Inconsistencies founded are mainly due to different criteria of analysis. In our paper, we took into account all identified RET + NSCLC patients, while Drilon et al. primarly focused on ITT-population. Anyway, we modified our manuscript in order to consider the same data as reported by Drilon et al.  

  • The authors clearly must add extra value to obtain a review that is distinct from Drilon 2020. It is worth highlighting that the authors raise unresolved questions in the discussion. However, although the authors list ongoing investigation attempting to answer these questions, they could be more critical and also propose ideas of their own. Furthermore, a section on the role of preclincial models and how the authors see that preclincial models could accelerate the development of a more optimal treatment of patients with RET gene alterations could be a way to improve and distinguish the authors review.

Answer: we thank the reviewer also for this comment. We improved the discussion section  by suggesting new ideas for subsequent future research in this field at the end of page 13 and 14: “Even though these trials are designed to assess the efficacy of anti-RET TKIs in a selected population, they will also demonstrate to what extent precision medicine techniques should be considered helpful in case of doubts. Despite the high reliability, some discrepancies might arise when comparing liquid and tissue biopsies, mainly due to high intra tumor heterogeneity. A possible implication of this concept, that could represent a further investigational area, would be to analyze the tumor heterogeneity among upfront non-responsive patients”.

Moreover, we added some preclinical evidences on the potential role of multiple TKIs, which mainly took into account combining non selective anti-RET TKIs with either CDK4/6 inhibitors or anti-PI3K/mTOR on line 16, page 14: “As previously described, concurrent genetic alterations have been identified in  tumor samples of RET rearranged NSCLC patients, mainly in TP-53 associated genes, cell-cycle associated genes and PI3Ksignalling pathway [6]. Furthermore, both in vitro and in vivo models of RET+ NSCLC cell lines demonstrated a strong synergistic effect when combining non-selective anti-RET TKIs with either CDK 4/6 inhibitors or PI3K/mTOR inhibitors [59]. In this scenario, available preclinical data highlight a potential role of multi-targeted agents in the treatment of RET-rearranged NSCLC, which is going to be further verified also in clinic. In an ongoing phase I trial including 13 RET + NSCLC patients, vandetanib plus everolimus led to 54% of ORR with important responses also seen in brain metastases. Median PFS was 4.4 months. Major toxiticies included diarrhea (21%), thrombocytopenia (16%), QTc prolongation (5%) and rash (5%) and 17/19 patients required dose reduction because of toxicities [60]. Taken together, all these data seems to demonstrate a possible crosstalk between different pathways both in vivo and in vitro. However,  given the predominant role of new selective anti-RET molecules, all these data might now be considered outdated. Thus, an interesting future field of investigation could be to further explore upfront combination strategy by using either pralsetinib or selpercatinib.   

  • The review has sections with many clinical trial numbers in the main text which is daunting to read and better suited for tables. This section could become more readable by providing more discussion on the meaning and critical assessment of the listed numbers in the main text.

 Answer: we thank the reviewer for this comment. We modified the discussion paragraph in order to streamline it.

  • Minor:100-102 is repeated text from abstract; L.188: AU: "Based on these evidences," → evidence is uncountable; L.201: AST not defined.

Answer: We thank the reviewer also for these comments. We modified the manuscript according to her/his suggestions.

Reviewer 2 Report

In this review Cascetta et al. describes the plethora of RET inhibitors focusing on current and future perspectives for RET positive NSCLC patients. The review is complete and well organized.

Author Response

In this review Cascetta et al. describes the plethora of RET inhibitors focusing on current and future perspectives for RET positive NSCLC patients. The review is complete and well organized.

Answer: we thank very much  the reviewer for the comments on our review.

Reviewer 3 Report

Dear authors, 

NSCLC is still a challenging disease. Oncogenic and druggable mutations are suitable for specific targeted treatments and hopefully, new drugs are on the horizon. New RET inhibitors are just about to be available for clinical use.   Therefore reviews on that topic are very welcome. The article is well structured, focusing on the distinctive patterns of RET + tumours, review of non-selective and selective RET inhibitors and future prospects/clinical trials.

Regarding the contents, it would be helpful to clearly distinguish among the non-selective inhibitors, which studies included RET patients and other settings.

Regarding Pralsetinb, the authors comment on the better activity in naive patients. That is true for ORR, but not DCR. Please check and reformulate. Regarding intracranial activity, if possible, please give more details, beyond good intracranial activity.

In table 1, please replace "pretreated or not" for pretreated or untreated, as in table. 

Finally, some suggestions, for language improvement:

controversy - controversial (line 132)

line 334: naive to treatments - treatment naive

avoid repeating importantly /more importantly

table 2: twicy - twice

line 437: on these bases - based on 

line 470- reformulate "...after they....tecniques. 

line 471 tecniques- techniques

Author Response

  • NSCLC is still a challenging disease. Oncogenic and druggable mutations are suitable for specific targeted treatments and hopefully, new drugs are on the horizon. New RET inhibitors are just about to be available for clinical use. Therefore reviews on that topic are very welcome. The article is well structured, focusing on the distinctive patterns of RET + tumours, review of non-selective and selective RET inhibitors and future prospects/clinical trials.

Answer: we thank the reviewer for her/his positive comments.

  • Regarding the contents, it would be helpful to clearly distinguish among the non-selective inhibitors, which studies included RET patients and other settings.

Answer: we thank the reviewer for this suggestion. We attempted to better elucidate among the non-selective inhibitors which trial included RET +  NSCLC patients.

  • Regarding Pralsetinb, the authors comment on the better activity in naive patients. That is true for ORR, but not DCR. Please check and reformulate. Regarding intracranial activity, if possible, please give more details, beyond good intracranial activity.

Answer: We thank the reviewer. We explained the differences observed between ORR and DCR in naïve patients. Furthermore, we added information on its intracranial efficacy.

  • In table 1, please replace "pretreated or not" for pretreated or untreated, as in table. 

Answer: Done

  • Finally, some suggestions, for language improvement: controversy - controversial (line 132); line 334: naive to treatments - treatment naïve; avoid repeating importantly /more importantly; table 2: twicy – twice; line 437: on these bases - based on; line 470- reformulate "...after they....tecniques. line 471 tecniques- techniques.

Answer: We thank the reviewer for all these comments. We modified the manuscript as suggested.  
